# Air Pollution Impairs Subjective Happiness by Damaging Their Health

**DOI:** 10.3390/ijerph181910319

**Published:** 2021-09-30

**Authors:** Yu Liu, Ke Zhu, Rong-Lin Li, Yang Song, Zhi-Jiang Zhang

**Affiliations:** 1College of Management, Wuhan Institute of Technology, Wuhan 430205, China; lyu429@163.com (Y.L.); zhuke24@163.com (K.Z.); ronglin722@163.com (R.-L.L.); summersong95@163.com (Y.S.); 2Department of Epidemiology, School of Public Health, Wuhan University, Wuhan 430071, China

**Keywords:** air pollution, residents’ happiness, health, mediating effect

## Abstract

*Background*: The impact of air pollution on residents’ happiness remains unclear and the underlying mechanism remains unknown. We aimed to examine the direct effect of air pollution on residents’ happiness and indirect effect through mediating their health. *Methods:* Based on the 2017 China Comprehensive Social Survey Data (CGSS), data on happiness were retrieved from 11,997 residents in 28 provinces in China. An ordered-probit model was used to examine the effect of air pollution on residents’ happiness and health, respectively. A stepwise regression was used to derive the direct effect of air pollution on residents’ happiness and indirect effect from health impairment attributable to air pollution. *Results:* Air pollution was associated with lower levels of health (coef. −0.190, 95% CI −0.212, −0.167, *p* < 0.001), while health was positively associated with happiness (coef. 0.215, 95% CI 0.196, 0.234, *p* < 0.001). Mediation analysis methods showed that air pollution impacted residents’ happiness directly and indirectly: the percent of total effect that was mediated through health was 36.97%, and the ratio of indirect to direct effect was 0.5864. *Conclusions:* Health plays a major mediating role in the relation between air pollution and residents’ happiness. In order to alleviate the impact of air pollution on residents’ happiness, future strategies should focus on health promotion besides reducing air pollutant emission.

## 1. Introduction

Happiness has become an important indicator reflecting the quality of people’s welfare [1]. A number of factors affect happiness, including income, health status and quality of living environment [2]. Air pollution, as the important factor affecting people’s happiness [3,4,5], has become an increasingly severe problem in recent years. According to State of Global Air in 2020 (a special report on global exposure to air pollution and its health impacts published by American Health Effects Institute), air pollution ranked 4th among major mortality risk factors globally, accounting for nearly 6.75 million premature deaths and 213 million years of healthy life lost [6].

The previous literature empirically demonstrated that air pollution had a negative impact on residents’ happiness [4,7,8,9,10]. Using the U.S. daily air pollution data, it was found that the increase in PM_10_ concentration significantly impaired residents’ happiness. The results of the study shown that the more serious the air pollution, the lower the subjective happiness of residents [4]. Based on repeated cross-sectional data from a representative, large-scale, social well-being survey of Mongolia, a negative relationship between air pollution and self-reported happiness was found [10]. A study found that air pollution had a significant negative impact on residents’ subjective happiness by matching micro-survey data on personal happiness and urban air pollution data. For every 1 ug/m^3^ increase in NO_2_ concentration, the reduction in residents’ happiness was probably 0.034% [7].

Most previous studies [4,7,8,9,10] focused on the impact of air pollution on residents’ happiness, but there are few studies on the mediating effect between air pollution and happiness. Mediation analysis is a statistical method commonly used to examine the interrelations among independent, mediating, and dependent variables to obtain the direct and indirect effects of an independent variable on a dependent variable [11,12]. Air pollution affected residents’ subjective happiness by stimulating residents’ senses [13]. Empirical results show that PM2.5 can significantly lower self-rated health and increase the probability of chronic diseases and mental depression [14]. Therefore, health may play a mediating role in the relationship between air pollution and residents’ happiness.

In general, a given variable may be said to function as a mediator to the extent that it accounts for the relation between an independent or predictor variable and a dependent or criterion variable [12]. Most studies on the mediating effect focused on psychology, exploring the effect of individual psychological perception on individual behavior [15,16,17]. There were also some mediating effect studies on subjective happiness, mainly in the field of sociology [18,19,20]. Few studies have examined whether health has a mediating effect in the relationship between air pollution and residents’ happiness. 

Accordingly, this study uses survey data and air pollution data to model individuals’ self-reported levels of “happiness” as a function of their demographic characteristics, mediating variables of “health”, and the air pollution at the place they were surveyed. Then, this study aims to explore the influence effect of air pollution on residents’ happiness, focusing on influencing effects of health as the mediating variable.

## 2. Method

### 2.1. Survey and Sampling

Data on residents’ happiness were obtained from the Chinese General Social Survey (CGSS) 2017. This survey was conducted by the Department of Sociology, Renmin University of China and Department of Social Sciences, aiming at comprehensively reflecting the transition of economics, politics, society, and culture in China. Its data are screened before publication in order to guarantee the reliability. CGSS data in 2017 were released in October 2020. The survey covered 478 villages in 28 provinces of China, encompassing 11,997 valid questionnaires. 

### 2.2. Model Specifications

Baron and Kenny method was used to verify whether health is the mediating variable, and analyze the influencing degree of the mediating variable [12]. Specifically, air pollution has a direct impact on residents’ happiness, i.e., the direct path. Health plays a mediating role between air pollution and residents’ happiness, i.e., the indirect path (Figure 1). 

The empirical model used in the present study was proposed by Baron and Kenny. Baron and Kenny proposed a 3-step approach to estimate and test the mediating effect [12]. The first step is to establish the regression model between the dependent variable (residents’ happiness) and the independent variable (air pollution), Model1. The second step is to establish the regression models between mediating variables (health) and independent variable (air pollution), Model2. The third step is to establish the regression model between dependent variable (residents’ happiness), independent (air pollution) and mediating variable (health), Model3. 

The ordered-probit model, first proposed by Mckelvey and Zavoina (1975) [21], has been widely used in discrete ordered variable estimation models (Wooldridge, 2002) [22]. The residents’ happiness is an ordered variable, and the ordered-probit model is usually used in empirical research. Therefore, we use the ordered-probit model to estimate the regression on the reference of Levinson’s study (2012) [4], Luechinger’s study (2010) [8], and Yang’s study (2014) [7].
$$Model1:Happiness_{ij}=\alpha_{1}\ln Pollution_{ij}+\beta_{1}Control_{ij}+\varepsilon_{1}$$
$$Model2:Health_{ij}=\alpha_{2}\ln Pollution_{ij}+\beta_{2}Control_{ij}+\varepsilon_{2}$$
$$Model3:Happiness_{ij}=\alpha_{3}\ln Pollution_{ij}+\gamma_{3}Health_{ij}+\beta_{3}Control_{ij}+\varepsilon_{3}$$
where $Happiness_{ij}$ is the dependent variable, denoting the happiness of the *j* resident of the *i* province, $Pollution_{ij}$ is the key explanatory variable, denoting the emission of air pollution in each province, $Health_{ij}$ represents the health of the *j* resident of the *i* province; $Control_{ij}$ is a vector composed of all of the control variables, which was derived from the literature on determinants of subjective happiness. It mainly includes the control variables such as gender, age, religion, marriage, housing area and respondents’ self-reported social equity. ε is the error term of the regression model, and α, β and γ are the regression coefficients of the equations. 

### 2.3. Dependent Variables

Residents’ happiness: We selected the 36th question in CGSS 2017: “Overall, do you think your life is happy?”. There are five choices for the respondents (very unhappy, unhappy, not possible to say happy or unhappy, happy, and very happy). The five choices were assigned values from 1 to 5, correspondently.

### 2.4. Independent Variables

Air pollution: Air pollution is defined as the presence of certain pollutants in the air at levels that have a harmful effect on human health and the environment [23]. SO_2_ emission is the main cause of the increase in PM_2.5_ [24], and is more irritating to human body [25]. At the same time, SO_2_ is the main taxable atmosphere pollutant in environmental protection tax regulations. Therefore, the SO_2_ emission was selected as a proxy variable for air pollution. The logarithmic transformation was used for SO_2_ emission. The emissions data on SO_2_ in 28 provinces were derived from the China Environmental Statistical Yearbook (2017).

### 2.5. Mediating Variables

According to the above literature review [14,26,27], air pollution was highly correlated with residents’ happiness, and health might be a mediator. Therefore, health in the present study was considered as mediating variable for observing the mediating effect of health on air pollution and residents’ happiness. We selected the 16th question in CGSS 2017: “The frequency of health problems affecting the work or daily activities over the past four weeks”. There are five choices for the respondents (always, often, sometimes, rarely and never). The five choices were assigned values from 1 to 5, correspondently.

### 2.6. Control Variables

Previous studies at the individual level show that gender, age, marriage, religion, house, social equity is closely correlated with residents’ happiness [28,29,30,31,32,33,34,35,36]. Therefore, Gender, Age, Age^2^, Religion, House, Marriage and Social equity were set as the control variables (Table 1). Gender and Religion were set as dummy variables. Considering the possible non-linear effect of the age of the residents on their happiness [28,29], we introduced the square term of residents’ age as the control variable [29]. Some studies have pointed out that housing area has become an important factor affecting individual’s happiness [30,31]. So, we introduced House, which represents the actual household housing area. Resident’s marriage status includes five parts: “Single, Married, Separated, Divorced, Widowed”. Additionally, Single is for reference only [32,33,34]. Social equity is an emotional experience generated by residents on the basis of value judgment [35]; it can not only promote the improvement of production efficiency, but also contribute to the improvement of residents’ happiness [36]. We selected the 35th question in CGSS2017 “Overall, do you think today’s society is fair?”. We defined it as respondents’ self-reported social equity. The value of five choices ranges from 1 to 5.

### 2.7. Descriptive Analyses

In the present study, age ranges from 21 to 106 years. The proportion of married residents was 77.1% (Table 1). The average household housing area of the respondents was 115.2 m^2^. The average value of happiness was 3.861. The respondents’ self-reported social equity was rated at 5 and the average social equity status was 3.103. The self-reported health was rated at 5 and the average health status was 3.902. The average SO_2_ emissions were 35.008 ten thousand tons. The proportion of male was 47.6% (Table 1), and the proportion of people who believe in religion was 10.3%.

## 3. Empirical Results

We used the STATA software to perform an ordered-probit regression for Model 1–3. The regression result was shown in Table 2. In the first step, for Model1, the air pollution regression coefficient was −0.097 (95% CI −0.120, −0.075) with *p* < 0.01. The emission of SO_2_ in air pollution had a significant negative impact on residents’ happiness. The statistical significance of control variables indicated that the effect of the regression analysis was robust. Females and married people have higher happiness level. The larger the housing area, the higher the sense of self-reported social equity, the more the happiness, which was consistent with previous studies [37]. The coefficient of Age was significantly negative, and the coefficient of Age^2^ was significantly positive, supporting the fact that there was a U-shaped curve relationship between age and happiness. The middle-aged people had the lowest happiness, and the result was basically similar to the previous studies [7].

In the second step (Table 2), the result of model 2 showed that the emission of SO_2_ in air pollution as an independent variable had a significant negative impact on health. The regression coefficient was −0.190 (95% CI −0.212, −0.167) with *p* < 0.01. The regression result indicated that the emission of SO_2_ significantly reduced residents’ health.

In the third step, the result of Model3 listed in Table 2. The coefficient of air pollution was −0.065 (95% CI −0.087, −0.042) and statistically significant (*p* < 0.01). Compared with results of Model1, the regression coefficient of air pollution on happiness was −0.097, revealing a significant mediating effect of health. In addition, the value of Pseudo R^2^ was 0.051 in model 1, and rose to 0.069 in model 3, which demonstrated that the effect of regression analysis was robust after the models contained the mediating variable.

In order to analyze the proportion of mediating effects in total effects, Sobel-Goodman Mediation test was estimated. The results of Sobel-Goodman Mediation test were listed in Table 3. The Z value was −12.80 with *p* < 0.01, indicating that the mediating effect of health was statistically significant. The Sobel-Goodman Mediation test results showed that the percent of total effect mediated by health was 36.97%. The ratio of indirect to direct effect was 0.5864.

## 4. Discussion

Based on the survey of 11,997 residents in 28 provinces in China, this study estimated the influence of air pollution on residents’ happiness, and explored the mediating effect of air pollution on residents’ happiness through health. We found that air pollution had significant negative effects on residents’ happiness. Meanwhile, the negative effects were divided into two parts: direct and indirect. One part was air pollution may directly affect residents’ subjective happiness. The other part was health played an important role in the indirect effect. The relationship between air pollution and residents’ happiness was influenced by the mediating effect of residents’ health.

One of the findings of this study is that air pollution is negatively associated with health, which is consistent with previous studies [27,38,39]. Low- and middle-income countries have experienced an intense process of urbanization and industrial development in a very short period of time, and this phenomenon has deleterious effects on the health of people resident in these developing countries [8]. Air pollution damages people’s health, increasing the risk of a variety of diseases [8,9]. Firstly, air pollution impacts residents’ psychological and mental health. Air pollution is associated with annoyance [40,41], anxiety [42] and mental disorders [43,44]. Even worse, air pollution may be a risk factor for cognitive functioning [45,46]. Secondly, long-term exposure to air pollution could cause chronic obstructive pulmonary diseases, lung cancer, cardiopulmonary mortality [27,47]. For example, fine particulate air pollution has been linked to coronary heart disease or cerebrovascular disease, coronary revascularization, myocardial infarction, and stroke [48]. Each 10-mug/m^3^ elevation in particulate matter was associated with approximately a 4%, 6%, and 8% increased risk of all-cause, cardiopulmonary, and lung cancer mortality [49].

Another one of the findings of this study is that health is positively related to happiness, which was basically similar to the previous studies [4,50]. For example, results from the study based on GSS (The General Social Survey, it is the full-probability, personal-interview survey designed to monitor changes in both social characteristics and attitudes currently being conducted in the United States.) Data showed that respondents with poor health reported average happiness of 1.7 on a 3-point scale; those with excellent health reported 2.4 [4]. According to the CGSS data, good health status had positive effect on happiness [8,51].

The most important finding of the present study is that health plays a mediating role on the relationship between air pollution and residents’ happiness. Most of previous studies set health as a control variable rather than a mediating variable [7,52]. This study explored the possible mediating effect of health in the relation between air pollution and residents’ happiness. The empirical results of present study showed that air pollution affected residents’ happiness through health. The proportion of mediating effect to total effect was 36.97%. This study provided new evidence for exploring the underlying mechanism for air pollution.

In terms of the mediating effect method, the stepwise regression was used in this study to analyze the relationship among mediating variables, dependent variables, and independent variables. Compared with the control variables, the stepwise regression method is helpful to describe the influence mechanism between variables clearly and specifically.

This study bears limitations. The present empirical analysis was based on the data of SO_2_ emission. Future research could be undertaken to collect the satellite based AOD data for each of the 28 provinces and then perform the same empirical analysis that had been carried out for SO_2_. Hence, adding AOD data and performing the analysis would strengthen the study. In addition, results of SO_2_ and AOD changes in different cities could be analyzed. In particular, the discussion on the changes of SO_2_ in the different provinces and its comparison globally, and the discussion of AOD at different sites and its comparison globally are meaningful. However, due to the limited data, we did not consider AOD changes in this paper.

## 5. Conclusions

This is the first study to provide clear evidence that benefits to residents’ happiness associated with air pollution are partially mediated through health. Better self-reported health was associated with higher happiness. In sum, the findings could provide reference to implement effective health protection policies in order to alleviate the impact of air pollution on happiness.

## Figures and Tables

**Figure 1 ijerph-18-10319-f001:**
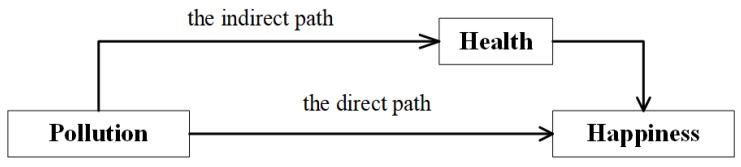
Mediating effect model of health.

**Table 1 ijerph-18-10319-t001:** Descriptive statistics.

	Variables	Measures	Mean/Proportion (%)	S.D.	Min	Max
Dependent variable	Happiness	The levels of respondents’ self-reported happiness (1–5)	3.861	0.846	1	5
Independent variable	Pollution	SO_2_ emissions (Ten thousand tons)	35.008	25.403	3.320	113.450
Control variables	Gender	Male = 1, Female = 0	47.6%			
	Age	The age of respondents	53.897	16.764	21	106
	Religion	No religion = 0, Religion = 1	10.3%			
	House	Household housing area (m^2^)	115.200	101.596	5	2400
	Social equity	The five levels of respondents’ self-reported social equity (1–5)	3.103	1.062	1	5
	Marital status					
	Single	Reference				
	Married	Married = 1, Others = 0	0.771	0.420	0	1
	Separated	Separated = 1, Others = 0	0.006	0.075	0	1
	Divorced	Divorced = 1, Others = 0	0.024	0.153	0	1
	Widowed	Widowed = 1, Others = 0	0.096	0.294	0	1
Mediating variables	Health	The levels of respondents’ self-reported health (1–5)	3.902	1.141	1	5

**Table 2 ijerph-18-10319-t002:** Impacts of pollution, health on the level of happiness.

		Model 1			Model 2			Model 3		
Dependent Variable		Happiness	95% CI	*p* Value	Health	95% CI	*p* Value	Happiness	95% CI	*p*-Value
Independent Variable	Pollution	−0.097	−0.120, −0.075	0.000	−0.190	−0.212, −0.167	0.000	−0.065	−0.087, −0.042	0.000
Control Variable	Gender	−0.094	−0.135, −0.054	0.000	0.137	0.097, 0.177	0.000	−0.125	−0.166, −0.084	0.000
	Age	−0.051	−0.059, −0.042	0.000	−0.033	−0.042, −0.025	0.000	−0.046	−0.055, −0.038	0.000
	Age^2^	0.000	0.000, 0.001	0.000	0.000	0.000, 0.000	0.030	0.000	0.000, 0.001	0.000
	Religion	0.009	−0.057, 0.076	0.780	0.075	0.010, 0.140	0.024	−0.003	−0.070, 0.063	0.921
	House	0.000	0.000, 0.001	0.001	−0.000	−0.000, 0.000	0.468	0.000	0.000, 0.001	0.000
	Social equity	0.317	0.298, 0.337	0.000	0.061	0.042, 0.080	0.000	0.312	0.292, 0.332	0.000
	Single	Reference								
	Married	0.355	0.271, 0.439	0.000	0.168	0.082, 0.255	0.000	0.330	0.245, 0.414	0.000
	Separated	−0.046	−0.321, 0.229	0.742	0.218	−0.060, 0.495	0.124	−0.091	−0.367, 0.185	0.519
	Divorced	−0.231	−0.380, −0.082	0.002	0.059	−0.092, 0.210	0.447	−0.249	−0.399, −0.100	0.001
	Widowed	0.058	−0.053, 0.169	0.306	0.029	−0.081, 0.139	0.607	0.055	−0.056, 0.167	0.332
Mediating Variable	Health							0.215	0.196, 0.234	0.000
Obs		11,997			11,997			11,997		
Pseudo R^2^		0.051			0.052			0.069		
Log likelihood		−12,924.977			−15,762.96			−12,678.482		

**Table 3 ijerph-18-10319-t003:** Sobel-Goodman mediation tests.

	Coef.	Std. Err.	Z	*p* > |Z|
Sobel	−0.0258	0.0020	−12.80	0.0000
Goodman-1	−0.0258	0.0020	−12.80	0.0000
Goodman-2	−0.0258	0.0020	−12.81	0.0000
Percent of total effect that is mediated: 36.97%				
Ratio of indirect to direct effect: 0.5864				

## Data Availability

Restrictions apply to the availability of these data. Data was obtained from National Survey Research Center at Renmin University of China and are available http://cgss.ruc.edu.cn/ (accessed on 13 September 2021) with the permission of National Survey Research Center at Renmin University of China.

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
