# Peer review of "Air Pollution Impairs Subjective Happiness by Damaging Their Health"

_ijerph, 2021, doi:10.3390/ijerph181910319_

Round 1

Reviewer 1 Report

The approach presented in the manuscript is novel, since most investigations on air pollution focus on tangible health effects and less on health related subjective effects. However, the significant contribution this peace of research can make is  mainly afected by unclear research design and presentation, description and discussion of results. 

I recommend the authors to clearly explain the research design in terms of the suitability of ordered-probit model to evaluate a five category dependent variable (happiness, Model 1) and a continious independent variable (SO2 emissions), as well as the dependent ordinal variable of health and the continious variable of SO2 emissions (Model 2); and lastly the five category dependent variable (happiness) and the independent variables of air pollution (continious) and health  (five categories) described in Model 3.

As far as know ordered-probit models apply to two category dependent variable in terms of one or more independen categorical variables which can have one or more ordered categories. 

Concerning the presentation of results, it is a mistake to describe a mean value and standard deviation  for a cualitative nominal variable, as it is the case of gender  and religion (table 1). In addition, I suggest to group together the variables: single, married, separated, divorsed and widowed, in a single ordinal variable labeled "social status" with five categories and from this you can show summary statistics.

I also suggest to find a better way to present the results in table 2.  Since from model specifications (lines 97-100) happiness, health and happiness are described as dependent variables in model 1-model 3 respectively. As I read results in table 2, I notice that listed variables under the first column headding are not the dependent variables according to describes models. 

Minor observations:

-What you mean by saying that "health is a transmission factor (line 22) 

-In model 2 air pollutin is an independent variable. However, you identify it as dependen variable (line 164)

-In line 193, specify what are those previous studies

-In line 205, specify what are those previos studies 

-References in lines 91 and 92 are not in the reference list

Author Response

The approach presented in the manuscript is novel, since most investigations on air pollution focus on tangible health effects and less on health related subjective effects. However, the significant contribution this peace of research can make is mainly afected by unclear research design and presentation, description and discussion of results. 

Response: We revised the manuscript and the quality has been improved greatly. We designed the study based on sufficient theoretical basis, selected appropriate data and used Stata software for empirical analysis, and discussed the similarities and differences between the conclusions of this paper and previous studies.

I recommend the authors to clearly explain the research design in terms of the suitability of ordered-probit model to evaluate a five category dependent variable (happiness, Model 1) and a continious independent variable (SO2 emissions), as well as the dependent ordinal variable of health and the continious variable of SO2 emissions (Model 2); and lastly the five category dependent variable (happiness) and the independent variables of air pollution (continious) and health  (five categories) described in Model 3.

As far as know ordered-probit models apply to two category dependent variable in terms of one or more independen categorical variables which can have one or more ordered categories. 

Response: We agree with the reviewer that Probit model can be applied for the modelling of two-category dependent variables. Meanwhile, ordered-probit model can be applied for the modelling of ordered dependent variables; in ordered-probit model, the independent variable can be category independent variable, continuous variable, or ordered variable. These principles have been recommended in the literature, e.g. R.D.McKelvey and W. Zavoina,”A Statistical Model for the Analysis of Ordinal Level Dependent Variables, ” Journal of Mathematical Sociology, vol.4,pp103-120, 1975.

Concerning the presentation of results, it is a mistake to describe a mean value and standard deviation  for a cualitative nominal variable, as it is the case of gender  and religion (table 1). In addition, I suggest to group together the variables: single, married, separated, divorsed and widowed, in a single ordinal variable labeled "social status" with five categories and from this you can show summary statistics.

Response: Thanks very much for your suggestions. We corrected the mean value to proportion (%) (table 1) for Gender and Religion. We summarized the five categories of single, married, separated, divorced and widowed under the variable “marital status”.

I also suggest to find a better way to present the results in table 2.  Since from model specifications (lines 97-100) happiness, health and happiness are described as dependent variables in model 1-model 3 respectively. As I read results in table 2, I notice that listed variables under the first column headding are not the dependent variables according to describes models. 

Response: We added a column named “Dependent variable” in table 2.

Minor observations:

-What you mean by saying that "health is a transmission factor (line 22) 

Response: Thanks very much for your suggestions. We corrected it tohealth might be a mediator.

-In model 2 air pollutin is an independent variable. However, you identify it as dependen variable (line 164)

Response: Thanks very much for your pointing out this mistake. We corrected it toindependent variable.

-In line 193, specify what are those previous studies

Response: We inserted citations here, i.e., ref 28, 39-40.

-In line 205, specify what are those previos studies 

Response: We inserted citations here, i.e., ref 5,51.

-References in lines 91 and 92 are not in the reference list

Response: We inserted citations here, i.e., ref 22,23.

Reviewer 2 Report

  • The authors have not sorted out for themselves (it seems) what they see as ‘happiness’ – they talk most of ten of ‘subjective’ happiness – this suggests to me there is also ‘objective’ happiness somewhere? If so – then need to clarify – if not – then why use ‘subjective’?
  • Many of the references are very old – i.e. the description of what the authors see as ‘happiness’ (P1L26) come from a very old and somewhat disconnected reference – linking happiness to economics - more recent references exists that will help to make more sense of this very subjective human condition. The authors need to make a good definition around more recent literature – such as the [4] reference.
  • P1L30 makes the statement ‘Air pollution, as the important factor affecting people’s happiness, has become a public concern’ makes no sense – it is a very important premise for this work but is not is referenced.
  • P1L31 refers to an unreferenced WHO report – this must be corrected
  • The authors do not clearly define what they mean ‘air pollution’ in the context of this submission – it rather appears they speak of polluted air – which is a different context to ‘air pollution’
  • Grammatical and syntactical editing required – some of the English is difficult to make sense of.
  • Reference [5] speaks of ‘satisfaction’ – which is not the same as the use of ‘happiness’ in this context.
  • P2L48 speaks of ‘but ignored the mediating effect’ – what mediating effect are we talking here ‘the effect’ implies one effect – but the authors do not specify. Only later in the paragraph does it become a bit clearer that ‘health’ is the probably mediator – although they are not emphatic about this. 
  • Authors are not specific about what of health – the paragraph P2L53-58 talks about studies that are more psycho-socio focussed – all of these are components of what we refer to as mental health – but they seem imply that their approach to health is different – in other words would not include these – what then DO they include in their definition of health?
  • P2L42-34 contains this statement … ‘Air pollution would increase the morbidity and mortality, such as respiratory diseases, cardiovascular disease’ - - they line is about disease (morbidity) when we are diseased – but we are not yet dead (mortality) – lumping these two words (M&M) into this sentence in this way make me think the authors have not thought through their definition of which part of health is dealt with in their work in reported in this submission. Therefore, we are not been clearly introduced to their work.
  • We also see the authors use ‘health’ and ‘subjective well-being’ (P2L61) – but if we look into the Methods – the mediating variable of ‘health’ is totally disconnected from this statement’.
  • The authors also do not give us any information about the control variables – why these were chosen and why they are important?
  • SO2 is not a good indicator for the purposes of this submission. Its application in terms of health effect is questionable.  Are there no direct PM5 data available in the same data set?  PM2.5 is a much broader pollutant group that is more often used as an indicator of health-related air quality.
  • ‘Transmission factor’ (P3L122) id not the correct expression. The authors speak of mediating throughout – which is the correct expression - why deviate?

Author Response

  • The authors have not sorted out for themselves (it seems) what they see as ‘happiness’ – they talk most of ten of ‘subjective’ happiness – this suggests to me there is also ‘objective’ happiness somewhere? If so – then need to clarify – if not – then why use ‘subjective’?

Response: In the field, happiness is termed as “subjective happiness”. There is no “objective happiness” though.

We added the description regarding variable “happiness” in 2.3. Dependent variables. The empirical data of happiness was from CGSS database. We selected the 36th question in CGSS 2017: “Overall, do you think your life is happy?”.

  • Many of the references are very old – i.e. the description of what the authors see as ‘happiness’ (P1L26) come from a very old and somewhat disconnected reference – linking happiness to economics - more recent references exists that will help to make more sense of this very subjective human condition. The authors need to make a good definition around more recent literature – such as the [4] reference.

Response: We agree with the review. We added description of happiness in our study in part 2.3. Dependent variables.

  • P1L30 makes the statement ‘Air pollution, as the important factor affecting people’s happiness, has become a public concern’ makes no sense – it is a very important premise for this work but is not is referenced.

Response: We modified the description here and added citations here, i.e., ref 4-6. The revised sentence is:

Air pollution, as the important factor affecting people’s happiness, has become an increasingly severe problem in recent years.

  • P1L31 refers to an unreferenced WHO report – this must be corrected

Response: Due to the change of WHO official website, this documentation is missing. Therefore, we deleted the WHO report and cited State of Global Air (2020), i.e., ref 7.

  • The authors do not clearly define what they mean ‘air pollution’ in the context of this submission – it rather appears they speak of polluted air – which is a different context to ‘air pollution’

Response: We added explanation for the variable ‘air pollution’ and inserted citation, ref 24.

Air pollution is defined as the presence of certain pollutants in the air at levels that have a harmful effect on human health and the environment, and SO2 emission is the main cause of the increase in PM2.5. Therefore, the SO2 emission was selected as a proxy variable for air pollution.

  • Grammatical and syntactical editing required – some of the English is difficult to make sense of.

Response: We corrected grammatical errors. We asked for colleague to proof read it. We believe the language in the revised manuscript does not constitute a problem for this study.

  • Reference [5] speaks of ‘satisfaction’ – which is not the same as the use of ‘happiness’ in this context.

Response: We agree with the reviewer. We changed the ref 5 to ref 11.

Sanduijav, C.; Ferreira, S.; Filipski, M.; Hashida, Y., Air pollution and happiness: Evidence from the coldest capital in the world. Ecological Economics 2021, 187, (3), 107085.

  • P2L48 speaks of ‘but ignored the mediating effect’ – what mediating effect are we talking here ‘the effect’ implies one effect – but the authors do not specify. Only later in the paragraph does it become a bit clearer that ‘health’ is the probably mediator – although they are not emphatic about this. 

Response: We added the explanation of mediating effect in L51 ‘Mediation analysis is a statistical method commonly used to examine the interrelations among independent, mediating, and dependent variables to obtain the direct and indirect effects of an independent variable on a dependent variable’, which was from the ref 12-13.

  • Authors are not specific about what of health – the paragraph P2L53-58 talks about studies that are more psycho-socio focussed – all of these are components of what we refer to as mental health – but they seem imply that their approach to health is different – in other words would not include these – what then DO they include in their definition of health?

Response: We added explanation for the variable health in 2.5. Mediating variables.

We selected the 16th question in CGSS 2017: “The frequency of health problems affecting the work or daily activities over the past four weeks”. There are five choices for the respondents (always, often, sometimes, rarely and never). The five choices were assigned values from 1 to 5, correspondently.

  • P2L42-34 contains this statement … ‘Air pollution would increase the morbidity and mortality, such as respiratory diseases, cardiovascular disease’ - - they line is about disease (morbidity) when we are diseased – but we are not yet dead (mortality) – lumping these two words (M&M) into this sentence in this way make me think the authors have not thought through their definition of which part of health is dealt with in their work in reported in this submission. Therefore, we are not been clearly introduced to their work.

Response: We deleted the term “mortality”.

We revised the statement to ‘Empirical results show that PM2.5 can significantly lower self-rated health and increase the probability of chronic diseases and mental depression’, and used new reference, i,e., ref 15.

  • We also see the authors use ‘health’ and ‘subjective well-being’ (P2L61) – but if we look into the Methods – the mediating variable of ‘health’ is totally disconnected from this statement’.

Response: We deleted the statement regarding ‘subjective well-being’.

  • The authors also do not give us any information about the control variables – why these were chosen and why they are important?

Response: Thanks for your suggestions. We added the reason of selection of control variables in 2.6 Control variables.

  • SO2 is not a good indicator for the purposes of this submission. Its application in terms of health effect is questionable.  Are there no direct PM5 data available in the same data set?  PM2.5 is a much broader pollutant group that is more often used as an indicator of health-related air quality.

Response: Thanks for your suggestions. There are no direct PM5 data available in the same data set or PM2.5 data in CGSS.

  • ‘Transmission factor’ (P3L122) id not the correct expression. The authors speak of mediating throughout – which is the correct expression - why deviate?

Response: Thanks for pointing out this inconsistency. We corrected it to ‘health might be a mediator’.

Round 2

Reviewer 2 Report

None to add

This manuscript is a resubmission of an earlier submission. The following is a list of the peer review reports and author responses from that submission.

Round 1

Reviewer 1 Report

This needs a full copy edit. There are incorrect use of phrases, and there are incorrect use of tenses (past and present). For example:

"Many literatures" is an incorrect phrase.

This is very distracting for the reader and it will need to be corrected in order for it to be read easily.

Methodology is described well. Conclusions support hypothesis and discussion. 

Reviewer 2 Report

The topic is very exciting and ambitious. Such studies especially from China, given its important role in battling the environmental crisis, can be valuable. Unfortunately, this paper fails to provide a clear narrative of the method and analysis. Overall the paper needs to be restructured and the method/data sections should be explained in a clear form.

Comments:

Line (41) you mention “Many literatures have carried out empirical research on this relationship” Please provide citations. Examples of the ones that have informed this research.

Line (43) citation for the study mentioned is missing. Is it reference number 5?  it is unclear.

2.2 model specification section seems to lack an introductory paragraph to explain the model choice.

Line (85) you mention “above hypothesis”: It might be helpful for the clarity of the paper to discuss the hypothesis in the instruction section, although it is discussed implicitly.

The rational for model choice is not clear.  You might consider restructuring the methods section and move the lines 116-119 to the beginning and expand it a little bit. It might be a good idea to talk about the methodological aspects of the previous studies in the introduction section and/or provide more information on the limitations of previous methods and how your approach might respond to some of the limitations. Overall the method choice and the method section are vague. 

2.3. Dependent variables: it is not in a narrative form. You are talking about the questionnaire, but it is vague and incomplete.

2.6 Control variable: By “fair” you mean “sense of social justice” that you previously mentioned in the introduction? It is not clear. Overall, I think you should provide more information on the questionnaire used in this research and maybe discuss some of the more abstract/composite factors such as social justice in more detail. Later (line 157) you talk about “sense of personal justice”, is it the same? Referring to previous studies that have used these variables can help make the method section clearer.

Creating a graph similar to the ones provided in SEM studies can be really helpful in navigating the analysis.  In the current draft information on different variables is incomplete and/or has been provided in bits and pieces across the introduction and the method sections.

Reviewer 3 Report

The study is important with respect to the advancements in understanding of air pollution and its impact on the society from a social science point of view. This study is important due to the interdisciplinary topics addressed, but needs some major modifications. Hence, this study should be considered for publication after addressing the following major comments.

Comments on introduction

The introduction needs a strong build up upon the context of air pollution in China. Better to provide trends of air pollution of major cities and then put forward why studying this topic is important from air pollution and happiness point of view particularly for China.

The authors need to pay attention on subscript and superscript notations e.g. PM10 instead of PM10 throughout the manuscript.

Comments on methods

Line no. 125: The authors suggest “Industrial SO2 emission is the main cause of PM2.5 climbing”. Sentence needs modifications with respect to English and grammar. Moreover, this sentence has to be backed up with a strong reference to support the claim that industrial SO2 increases PM2.5.

Additionally, it would be important to see if the authors extracted freely available satellite based AOD data for each of the 28 provinces and then perform the same exercise that has been done for SO2. This would make the results concrete.

Presently the results are based on only industrial SO2. Hence, adding AOD data and performing the analysis would strengthen the study.

Comments on results

It would be nice to see the results of industrial SO2 changes in different cities, in the beginning of this section that would indicate it is a bigger problem in the country.

Then once the authors perform an analysis with AOD that result should also show up here.

This would be followed by the empirical results for both SO2 and AOD.

Comments on discussions and conclusions

This section lacks a strong discussion with previous literature.

There is no discussion on the changes of industrial SO2 in the different provinces and its comparison globally.

Once the authors perform analysis of AOD, discussion of AOD at different sites and its comparison globally would add value to the manuscript.

Finally, how changes in industrial SO2 and AOD could impact health and thus adversely impact happiness using empirical equations and its discussion with existing trends of health and happiness would be nice.